# Temporal variations of groundwater table and implications for submarine groundwater discharge: A three-decade case study in Central Japan

Bing Zhang[1, 2, 3], Jing Zhang[2*, 1], Takafumi Yoshida[1]

[1] Northwest Pacific Region Environmental Cooperation Center, 930-0856, Toyama, Japan;
[2] Earth and Environmental System, Graduate School of Science and Engineering, University of Toyama, 930-8555, Toyama, Japan
[3] Tianjin Key Laboratory of Water Resources and Environment, Tianjin Normal University, 300387, Tianjin, China

* *Correspondence to*: Jing Zhang (jzhang@sci.u-toyama.ac.jp)
Tel: +81-76-445-6665 Fax: +81-76-445-6549

**Abstract.**

Fresh submarine groundwater discharge (SGD) is the key pathway of flux and nutrients for the groundwater from land to the ocean. SGD flux is a current issue of discussion and a means to clarify the coastal marine system under climate change. SGD flux accounts for about 1/4 of the river runoff in the Katakai alluvial fan in Uozu, Toyama, Japan, which is an ideal area to study SGD flux considering the need for a rapid response to climate change and the prior research on SGD there. In this paper, the monthly groundwater table's condition over 30 years is analyzed using monthly rainfall, snowfall, and the climate change index. Rainfall has been on an upward trend, but the snowfall has decreased over 40 years. Furthermore, the groundwater table at monitoring wells in the coastal area increased, as a result of the increased rainfall. However, the relationship between snowfall and groundwater is negative. As expected by Darcy's law, SGD flux was controlled by the hydraulic gradient of the coastal groundwater. The estimated historic SGD flux by groundwater table variation shows an upward trend of SGD. Considering the increase in precipitation and the groundwater table, SGD flux may increase under climate change.

## 1 Introduction

Fresh submarine groundwater discharge (SGD) is the direct flow of groundwater into the ocean. The groundwater flows down a gradient, and SGD occurs wherever a coastal aquifer is connected to the sea (Chen et al., 2005; Zhang and Mandal, 2012). SGD has been recognized as not only an important source of freshwater discharge into the ocean but also a valuable component of the hydrological cycle between the terrestrial groundwater system and the marine environment

(Church, 1996; Taniguchi et al., 2002; Hatta and Zhang, 2013; Liu et al., 2014). The estimation of global SGD varies from 0.2 % to 10 % of the river flow (Burnnet et al., 2001). SGD may be both volumetrically and chemically important to coastal water and chemical budgets (Taniguchi et al., 2002). Thus, an accurate estimate of SGD flux is essential to predict future coastal environment under climate change conditions.

5         Toyama Bay is an ideal site to study SGD because it is easily accessible and has been studied in previous reports on several SGD sites off the coastal area (Fujii et al., 1986; Zhang and Satake, 2003; Nakaguchi et al., 2005; Hatta and Zhang, 2013). The SGD flow rates in the Katakai Alluvial Fan (~680 m/yr; (Zhang et al., 2005)) are greater than most of those reported worldwide (~454 m/yr; (Taniguchi et al., 2002)). The seepage water collected off Uozu is the potential type, which is the flow of the groundwater controlled by the potential of the spring in terms of conversion points of the geomorphologic

gradient (Zhang and Satake, 2003). The source of the freshwater is the precipitation in the Toyama region. Furthermore, this area has shown a rapid response to climate change, with a recent reduction in the year-round snow extent on the top of nearby mountains, which is shown here to affect SGD flux into the coastal environment (Hatta and Zhang, 2013).

        Groundwater is the linkage between precipitation and SGD. In this paper, we describe the temporal variations in the monthly groundwater table, rainfall and snow from 1985 to 2015. Furthermore, we analyzed the relationship between

rainfall, snowfall, and El Niño and La Niña events. The relation between SGD flux and the groundwater table was also studied. Finally, we discuss the impact of climate change on rainfall, the coastal groundwater system, and SGD.

# 2 Study Area

## 2.1 Study site description

        Toyama Bay, a semi-enclosed bay in central Japan, connects to the Sea of Japan at its northern boundary. There is

buried forest and submarine groundwater seepage off Uozu, at the east side of Toyama Bay (Zhang and Satake, 2003). The Katakai River alluvial fan is located at Uozu, Toyama prefecture (Fig. 1). The SGD area of the Katakai River alluvial fan occurs at 150–200 m seaward of the coastline in water depths of 8 and 22 m. The average fluxes were 0.8–1.3 L/min/m$^2$ at 8 m and 0.5–0.8 L/min/m$^2$ at 22 m from April to December 2003 (Zhang et al., 2005).

        The Katakai River alluvial fan is a coastal well watered fan into which the Katakai River is deposited. The total area of

the Katakai River catchment is 169 km$^2$. The length of the main river is 27 km from the water source on Kekachi Mountain (2414 m). The average slope of the Katakai river bed is 8.5 %, which is the most steeply sloped river of the seven rivers in Japan. The average flux of the Katakai River is 10.2 m$^3$/s. Annual precipitation is 2500 mm in Uozu, and it is about 4000 mm in the mountainous area. The annual average temperature is 14 $^o$C. The annual potential evapotranspiration is 765 mm.

Figure 1 Locations of groundwater wells in Katakai River alluvial fan

## 2.1 Hydrogeology setting

The groundwater head at the top of the alluvial fan is about 120 m (Fig. 1). The groundwater head gradient is high, due to the slope topography. The aquifer of the Katakai river fan is gravel and sand sediments. There are four ancient river courses in the central fan. The hydrogeological setting of Uozu consists of three layers. Layer A (top layer) is alluvium at the Holocene, consisting of gravel, sand and clay layers. The thickness of layer A from the hill area to the coastal fan area is from 20 m to about 100 m, respectively. Layer B (middle layer) is the deposit of the dissected fan at the late and middle Pleistocene, including gravel, sand and clay layers. The thickness of layer B in the fan area is about 80 m. Layer C (bottom layer) is the deposit of the dissected fan at the early Pleistocene. The bedrock is sandstone and mudstone (Kokusai Kogyo Co. Ltd., 2002). There are artesian wells along the coastal area, due to the existence of clay layers in the sand aquifer.

## 3 Methods

### 3.1 Data source

The groundwater table monitoring data (from 1985 to 2015) was obtained from the Water Information Database, Uozu, Toyama Prefecture, Japan. The precipitation data (from 1976 to 2015) and historical El Niño/La Niña events were obtained from the website of the Japan Meteorological Agency (http://www.jma.go.jp/jma/index.html). The oceanic Nino index (ONI) was from the Center for Weather and Climate Prediction (CPC), National Oceanic and Atmospheric Administration (NOAA), U.S. Department of Commerce. The ONI values are calculated by the monthly Nino 3.4 index. The statistical characteristics and linear regression of the monthly groundwater table, rainfall and snowfall were analyzed by SPSS software.

### 3.2 Analytical methods

Wavelet transforms are a very powerful tool in which to analyze non-stationary signals. It allows for the identification of the main periodicity in a time series and the evolution at the time of each frequency (Liang et al., 2011). The cross wavelet transform is used to examine relationships in time frequency space between two time series (Labat, 2010). Phase angle statistics can be used to gain confidence in causal relationships between the time series (Grinsted et al., 2004; Zhang and Wang, 2016).

The methods of continuous wavelet transform (CWT) and cross wavelet transform (XWT) provide the basis for wavelet coherence analysis (Grinsted et al., 2004). The wavelet coherence (WTC) of two time series was defined as:

$$R_n^2(s) = \frac{\left| S(s^{-1} W_n^{XY}(s)) \right|^2}{S(s^{-1} \left| W_n^X(s) \right|^2) \cdot S(s^{-1} \left| W_n^Y(s) \right|^2)} \ , \tag{1}$$

where $S$ is a smoothing operator. The wavelet coherence is a localized correlation coefficient in time frequency space (Grinsted et al., 2004). The arrows → and ← in the WTC figures indicate the positive and negative relationship between two time series, respectively. Meanwhile, the arrows ↓ and ↑ show that time series 1 is one-quarter period earlier and later, respectively, than time series 2 (Zhang and Wang, 2016).

# 4 Results

## 4.1 Groundwater table variation

There are six groundwater monitoring sites, including eight monitoring wells in the Katakai river alluvial fan (Table 1). The average groundwater table decreased from the mountain area to the coastal plain area; the groundwater table of Rokuromaru is the highest, with the average values of 42.72 m and 40.96 m, respectively. The groundwater of Kichijima (24.96 m) is the second highest, followed by that of Higashiosaki (19.77 m). The groundwater tables of Shinkanaya and Kyoden are about 10 m. The groundwater level of Kitaonie, which is the nearest to Toyama bay, is the lowest (6.83 m). The range variations of Rokuromaru and Kichijima are the largest, while that of Kyoden is the smallest. The standard deviation of shallow groundwater is larger than the deep groundwater at the same site.

Table 1 Description of groundwater monitoring wells

The groundwater table trends of the monitoring wells are similar (Fig. 2). Linear regression was applied to analyze the trends of the groundwater variation. The standardized coefficients of most groundwater wells are significantly positive, indicating an increase in groundwater table (Table 1). However, the groundwater table of Kichijima may decline, since the standardized coefficient is significantly negative.

Figure 2 Monthly groundwater table, rainfall and snowfall from 1985 to 2014 (a), from 2010 to 2014 (b)

## 4.2 Rainfall and snowfall variations

The average rainfall and snowfall in Uozu was 2473 mm and 385 cm, respectively, from 1976 to 2015 (Table 2). However, rainfall increased while snowfall decreased during these years. Rain increased by 14.76 % and snowfall decreased by 26.94 % over the past 20 years, compared to the 20 years from 1976 to 1996. The linear regression of annual rainfall is y=18.322x -34088.74 (p<0.01), indicating increased rainfall in the future. However, the annual snowfall amount decreased, with a linear regression of y=-6.712x+13792.71 (p<0.05). Furthermore, the percentage of snow in total precipitation was 19.44 % from 1976 to 1996, and this declined to 12.38 % from 1997 to 2015.

Table 2 Description and prediction of rainfall, snow and water budget

## 4.3 Relationship between rainfall, snowfall and groundwater

The groundwater table in summer (June to August) is the highest, while the groundwater table in early spring (March to May) is the lowest (Fig. 2 a, b). The groundwater table declined sharply during winter, especially after snowfall. The groundwater table is the lowest one or two months after the end of snowfall. Comparing the peaks of rainfall and the groundwater table, the rainfall in August 2010 was the largest, and the highest groundwater table occurred two months later. The groundwater table also increased to its peak two months after the peak rainfall in December 2012 and August 2013 (Fig. 2 b).

Figure 3 Squared wavelet coherence between rainfall and groundwater table (a), snowfall and groundwater table (b), rainfall and ONI (c), snowfall and ONI (d)

The relationships among rainfall, snowfall and groundwater table was analyzed by wavelet coherence. Taking the Shinkanaya (30 m) monitoring well as an example, the relationship between groundwater table and rainfall was positive from 1990 to 2010, with a period of 2–4 years (Fig. 3a), while the groundwater lagged behind by about a quarter year from 1985 to 1995, 1998 to 2002, and 2003 to 2015, with a 1-year period. However, the groundwater table is negatively correlated with snowfall from 1985 to 2015, with a 1-year period (Fig. 3b). This result coincides with that in Fig. 2b. The groundwater table decreased when snowfall begin in winter.

## 4.4 Relationships among rainfall, snowfall and climate change index

Wavelet coherence was used to analyze the relationships among rainfall, snowfall and climate change index (ONI). The relationship between rainfall and ONI is significantly negative in 1- and 3–5-year periods (Fig. 3c), indicating that the La Niña events may increase rainfall. The climate change index was about a quarter year earlier than snowfall beginning in a 0.5-year period (Fig. 3d). The climate change index, associated with El Niño and La Niña events, significantly influences rainfall and snowfall.

For the analysis of the relationship between climate change and rainfall, snowfall is the basis used to determine the impact of climate change on groundwater. We used the Oceanic Niño Index (ONI) to estimate the climate changes associated with the El Niño and La Niña events.

Figure 4 Rainfall (P), snowfall (Snow), oceanic Nino index (ONI), El Niño and La Niña events

There were six El Niño and six La Niña events from 1985 to 2015 (Fig. 4). The seasonal rainfall and snowfall during El Niño events were 642 mm and 155 cm, and those during La Niña events were 635 mm and 157 cm, respectively. The ratios of snowfall and rainfall during El Niño and La Niña events were 1.17 and 1.39, respectively. The most extreme El Niño event was from spring 1997 to spring 1998 (ONI=2.3). The most extreme La Niña events occurred during spring 1988 to spring 1989 (ONI=-1.7). The La Niña events may have caused more snowfall and more extreme monthly snowfall than El Niño.

## 4.5 Relationship between groundwater and SGD

The submarine groundwater discharge off Uozu is controlled by the potential of the geomorphologic gradient (Zhang and Satake, 2003). The Darcy's law describes the water flow through a porous medium (sand). The groundwater aquifer in Uozu is sand. Thus, according to Darcy's Law, the groundwater flow rate is correlated with the hydraulic gradient (Mulligan and Charette, 2006). The SGD flux is positive for the groundwater table. Using the monitoring SGD flux over April to August 2003 (Zhang et al., 2005), we established the relationship between monthly SGD flux ($y$, m/month) and groundwater variation ($x$, m/month).

Here, $x$ is the summarized daily groundwater table variation (daily groundwater table above 5.5 m).

For SGD flux at 8 m, $\qquad y = 0.45x + 25.28$ (R$^2$= 0.819) $\qquad$ (2)

For SGD flux at 22 m, $\qquad y = 0.65x - 2.53$ (R$^2$= 0.819) $\qquad$ (3)

The monthly SGD flux may be estimated according to the monthly groundwater table variation. Then, the annual SGD flux can be calculated based on the monthly SGD flux. The estimated SGD flux off the Katakai river alluvial fan is shown in Figure 5. The estimated data shows that the SGD flux is dominated by the groundwater table variation. Furthermore, SGD flux increased as a consequence of the groundwater table increasing. The SGD flux we estimated is 8 m and 22 m off Uozu. However, the fresh SGD flux at 40 – 100 m is about 2 to 4 times of 0 – 40 m (Hatta and Zhang, 2013). Compared to the results of water budget (Table 2), the fresh SGD maybe underestimated. However, since the groundwater table is easy to determine, the fresh SGD flux could be estimated by the equations 2 and 3 in the coastal sand aquifer.

Figure 5 Estimated SGD flux by groundwater table

## 5 Discussion

## 5.1 Impact of climate change on rainfall

The annual precipitation at Uozu, Toyama prefecture is about 2500 mm, and it's in an increasing trend. The precipitation is 1.52 times the mean precipitation (1634 mm) in Japan (Xu et al., 2003). This high precipitation is caused by high seawater temperature, which is maintained by the warm Tsushima current moving at 2.6 million m$^3$/s from the southwest to the northeast along the Japan Sea coast (Zhang and Satake, 2003). The cold northwesterly wind gathers much water vapor from the Sea of Japan, bringing heavy snow to the Sea of Japan side (Kawase et al., 2013). Furthermore, the precipitation patterns are dominated by shifts as sea-surface temperatures change, e.g. El Niño and La Niña (Trenberth, 2011). Climate change, with warming conditions and increased moisture, may produce more intense precipitation events. More precipitation occurs as rain instead of snow, and snow melts earlier (Emori, 2005; Trenberth, 2011; Fischer and Knutti, 2015). The impact of climate change on precipitation is changing the precipitation amount and type, causing an increase in precipitation (about 0.1 to 24.5 mm/decade) (Wang et al., 2017) and a decrease in snow at Toyama. The annual precipitation may increase to about

3000 mm by 2030, and annual snowfall will decrease to less than 1000 mm by 2050 (Table 2). Moreover, the El Niño and La Niña events influence regional anomalous circulation features (Leung et al., 2017), and the frequency of extreme precipitation.

## 5.2 Implication for coastal groundwater system

The groundwater table variation is not only determined by precipitation but also by human activities. The utilization of snowmelt for groundwater may cause the lowest groundwater table in winter at Uozu (Kokusai Kogyo Co. Ltd., 2002). Moreover, climate change in combination with increased anthropogenic activities will affect coastal groundwater systems (Essink et al., 2010). In a warmer world, less winter precipitation falls as snow, and the melting of winter snow occurs earlier in spring. Meanwhile, much of the winter runoff will immediately be lost to the oceans (Barnett et al., 2005). Furthermore,
the groundwater extraction may cause groundwater depletion. The contribution of groundwater depletion to global sea-level rise amounted to 0.27 mm/yr in 2000 (Wada et al., 2016). The impact of groundwater extraction on coastal aquifers was more significant than the impact of sea-level rise in groundwater recharge (Ferguson and Gleeson, 2012). Due to the increased precipitation, the groundwater recharge amount increased in Uozu. With snowfall decreasing, the portion of groundwater provided by snowmelt may decline. Furthermore, the El Niño and La Niña events change the groundwater table
pattern. The El Niño events increase the groundwater table in winter, while the La Niña events increase the groundwater table sharply in summer (Fig. 2). Under climate change and the influence of human activities (Han et al., 2016), the groundwater table may increase with an irregular pattern in the future.

## 5.3 Implication for submarine groundwater discharge

    According to the terrestrial water budget, an estimated $33 \times 10^8$ m$^3$/yr of groundwater discharged from the continental
shelf into Toyama Bay as fresh submarine groundwater discharge (Ito and Fuji, 1993; Zhang and Satake, 2003). The estimated submarine groundwater discharge is approximately $6 \times 10^7$ m$^3$/yr, which is 20 % of the river's outflow ($30 \times 10^7$ m$^3$/yr) in Uozu (Kokusai Kogyo Co. Ltd., 2002) (Table 2). However, the precipitation varies under climate change, because the ratio of rainfall and snow would increase. Furthermore, the variations of meteorological parameters, e.g. temperature, humidity, may cause the changes of evapotranspiration (Cong et al., 2009; Shimizu et al., 2015), as well as river runoff,
groundwater discharge in the water budget. Thus, the uncertainty of percentage of evapotranspiration, river runoff and groundwater discharge in the total water budget may exist.

    SGD is one of the indicators that reflect the effects of climate change on the marine ecosystem. It has been reported that the SGD-sea level correlation was high (Taniguchi and Iwakawa, 2004). However, the increased head in the groundwater system at the coast can be easily produced, due to the highly permeable Holocene and Pleistocene layers
(Kokusai Kogyo Co. Ltd., 2002; Essink et al., 2010). The average concentration of $NO_3^-$ in fresh SGD (0.69 mg/L) is larger than riverine input (0.18 mg/L) (Hatta and Zhang, 2013). The estimated SGD flux is described in Table 2 and Figure 5. The

SGD flux increase over three decades, the DIN flux in the SGD to Toyama bay may increase. Thus, the submarine groundwater discharge is a significant source of nutrition, more than river water, to the coastal marine ecosystem of Toyama Bay (Zhang and Satake, 2003; Lee et al., 2010).

5     Moreover, SGD flux may affect the availability of planktonic food for fish larvae. Fish production appears to be controlled by the climatic factors governing the processes in upwelling systems (Walther et al., 2002). Due to the increased precipitation and groundwater table, SGD flux may increase under climate change in the future. However, the annual SGD flux may be around $7 \times 10^7$ m$^3$ and $8 \times 10^7$ m$^3$ by 2030 and 2050, respectively (Table 2), as a result of the increase in the groundwater table (Table 3). The increased amount of SGD is less than river runoff, since most increased precipitation changes into river runoff to the ocean.

# Conclusions

Groundwater table, rainfall, and snowfall datasets from 1985 to 2015 were collected to analyze their variations. The relationships among groundwater table, rainfall, snow, and climate change events were analyzed by wavelet coherence to discuss the implications for climate change. The results are summarized as follows:

(1) The groundwater table is the highest in summer and the lowest in spring. The average groundwater table decreased 15     from the mountainous area to the coastal plain area. Linear regression reflected the increase in the groundwater table in almost all monitoring wells. Rainfall increased and snowfall declined over 40 years.

(2) The relationship between the groundwater table and rainfall is positive. The groundwater tables increased to the peak one to two months after the peak rainfall. The groundwater table is negatively correlated with snowfall. The climate change index associated with El Niño and La Niña events, especially La Niña, may cause extreme rainfall and snowfall.

20     (3) SGD flux was controlled by the hydraulic gradient of the coastal groundwater. The linear regression between SGD flux and the groundwater table was established. The historic SGD flux was estimated by groundwater table variation. The upward trend of the precipitation and groundwater table may indicate an increase in SGD flux although with some uncertainty.

This study demonstrates that groundwater is the linkage between climate change and submarine groundwater discharge 25     with long time datasets. Due to increases in precipitation and the groundwater table, the flux of submarine groundwater discharge will increase under climate change. In addition, the quality of submarine groundwater discharge should be clarified under climate change conditions worldwide in the future.

# Acknowledgements

This research was supported by the Grant‐in‐Aid for Scientific Research on Innovative Areas Grant (25110505 and 15H00973), the JSPS KAKENHI Grants (JP26241009) and the Environment Research and Technology Development Fund (S-13) of the Ministry of the Environment, Japan.

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

Table 1 Description of groundwater monitoring wells

| No. | Well | Depth m | Screen depth m~m | Data period (years) | Groundwater table (Mean±SD) m | Linear regression* |
|---|---|---|---|---|---|---|
| 1 | Shinkanaya | 100 | 72-94 | 1985--2015 | 9.71±0.75 | y=0.01x-4.75 (p=0.12) |
| 2 | Shinkanaya | 33 | 17-28 | 1985--2015 | 9.69±0.77 | y=0.12x-14.99 (p<0.01) |
| 3 | Kichijima | 80 | 25-36 | 1985--2015 | 24.96±2.44 | y=-0.07x+171.88 (p<0.01) |
| 4 | Higashiosaki | 42.5 | 9-20 | 1985--2015 | 19.77±1.32 | y=0.04x-63.77 (p<0.01) |
| 5 | Kyoden | 100 | 56-67, 78-89 | 1985--2015 | 9.86±0.38 | y=0.01x-7.30 (p<0.01) |
| 6 | Rokuromaru | 38 | 27-33 | 2004--2015 | 42.72±4.08 | y=0.41x-780.20 (p<0.01) |
| 7 | Rokuromaru | 80 | 64-75 | 2004--2015 | 40.96±4.01 | y=0.36x-683.24 (p<0.01) |
| 8 | Kitaonie | 70 | 59-71 | 2002-2015 | 6.83±0.49 | y=0.04x-76.76 (p<0.01) |

* Linear regression between groundwater table (y) and month (x)

Table 2 Description and prediction of rainfall, snow and water budget

| | Average (Mean±SD) mm/yr | | Water budget* $10^7\,m^3/yr$ | | | | |
|---|---|---|---|---|---|---|---|
| | rainfall | snowfall | Precipitation | Evapo-transpiration | River runoff | Groundwater usages | SGD |
| 1976~1996 | 2311±616 | 4492±2629 | 47±13 | 11±2.9 | 28±7.5 | 2.0±0.5 | 6.0±1.6 |
| 1997~2015 | 2652±300 | 3282±1346 | 54±6.1 | 13±1.5 | 32±3.6 | 2.0±0.2 | 6.0±0.68 |
| 1976~2015 | 2473±516 | 3850±2110 | 50±10 | 12±2.5 | 30±6.3 | 2.0±0.4 | 6.0±1.2 |
| 2010~2030 | 2949±150 | 2573±987 | 60±3.1 | 14±0.73 | 36±1.8 | 2.4±0.1 | 7.2±0.37 |
| 2030~2050 | 3147±695 | 970±387 | 64±14 | 15±3.4 | 38±8.5 | 2.6±0.6 | 7.7±1.7 |

*Water budget is calculated by percentage of evapotranspiration (24%), river runoff (60%), groundwater usages (4%) and submarine groundwater discharge (SGD, 12%) to precipitation from 1976 to 2015 in Uozu.

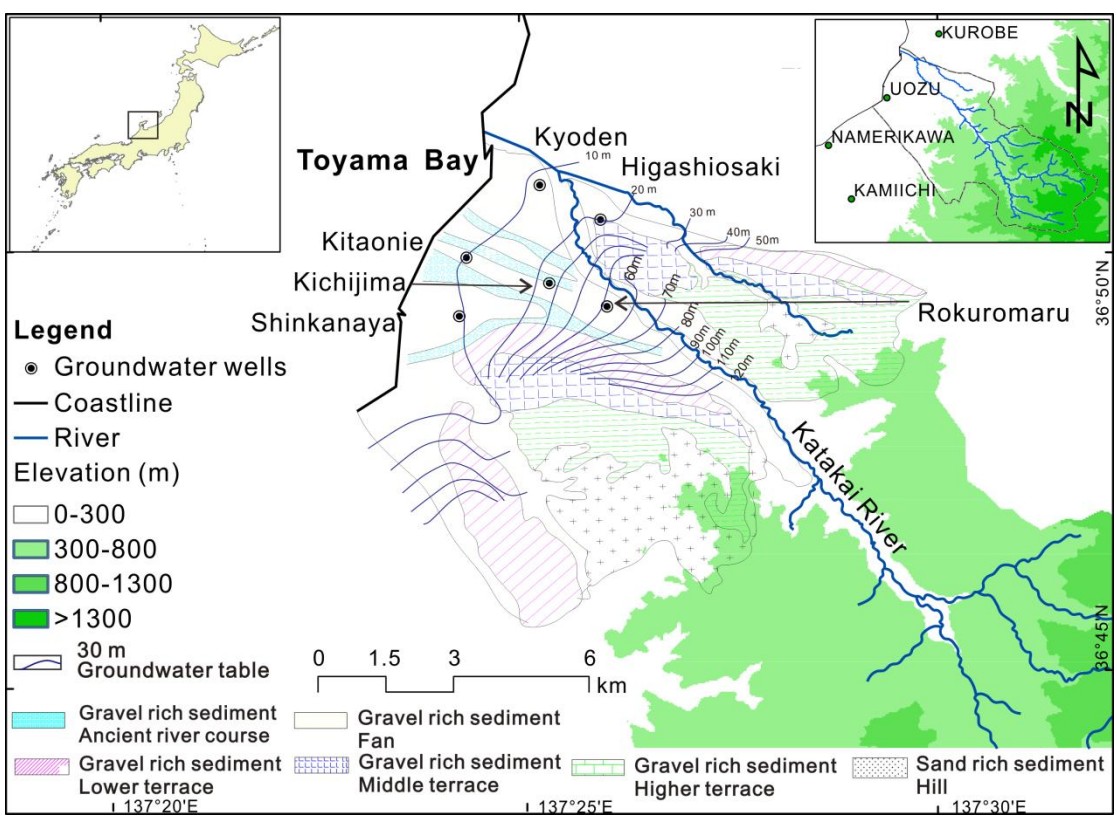

Figure 1 Locations of groundwater wells in Katakai River alluvial fan

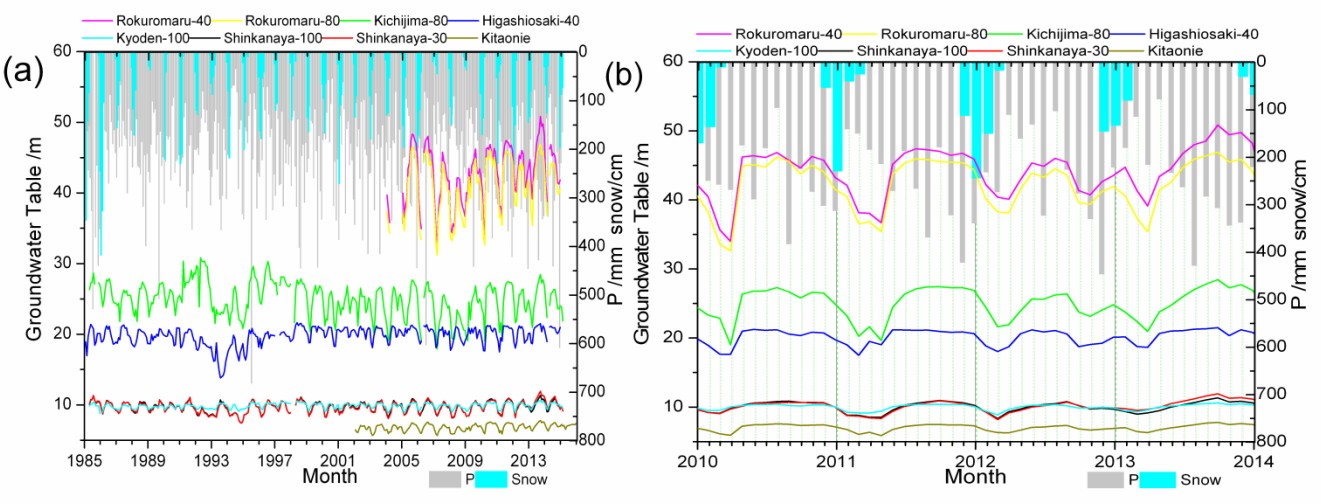

Figure 2 Monthly groundwater table, rainfall and snowfall from 1985 to 2014 (a), from 2010 to 2014 (b)

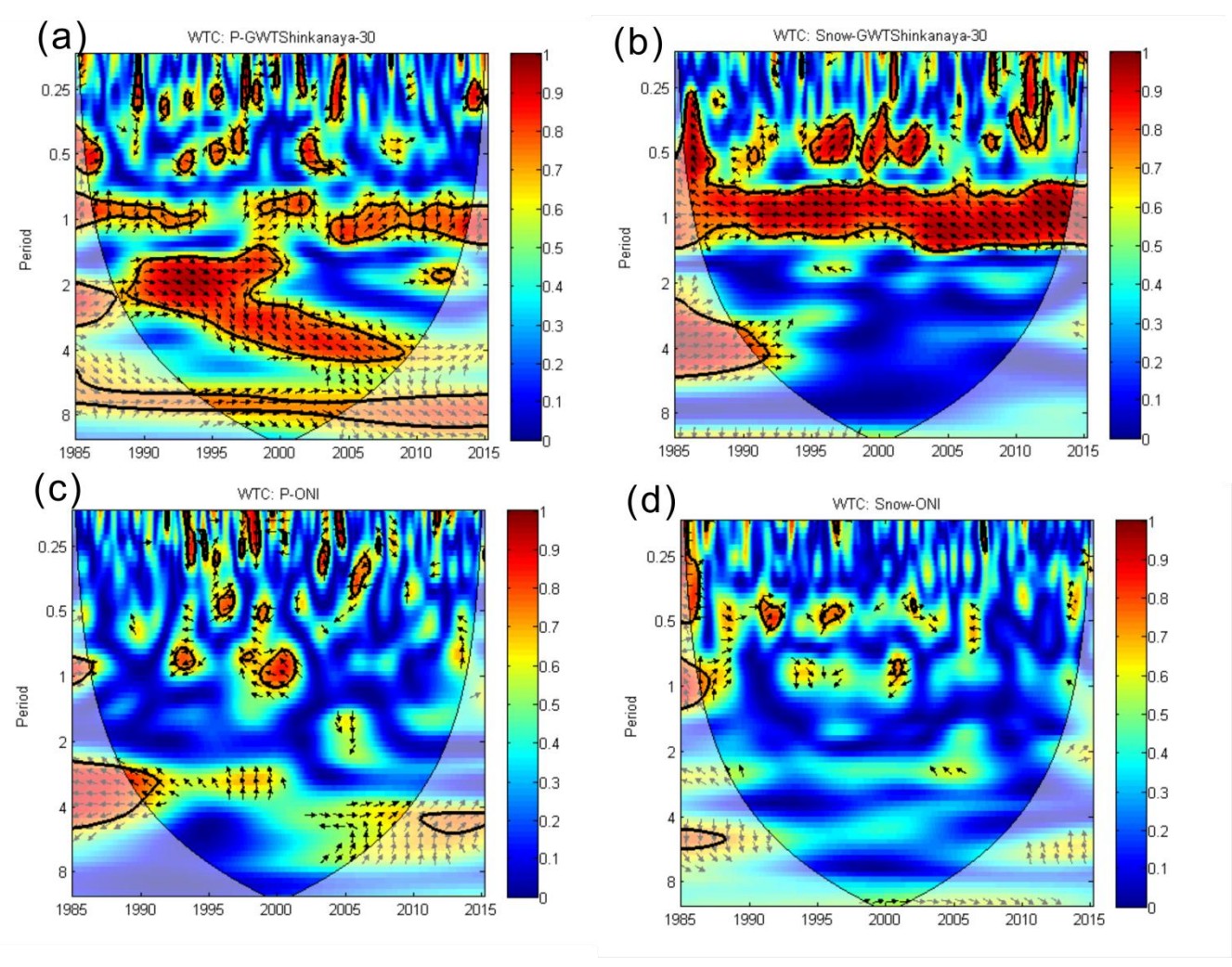

Figure 3 Squared wavelet coherence between rainfall and groundwater table (a), snowfall and groundwater table (b), rainfall and ONI (c), snowfall and ONI (d)

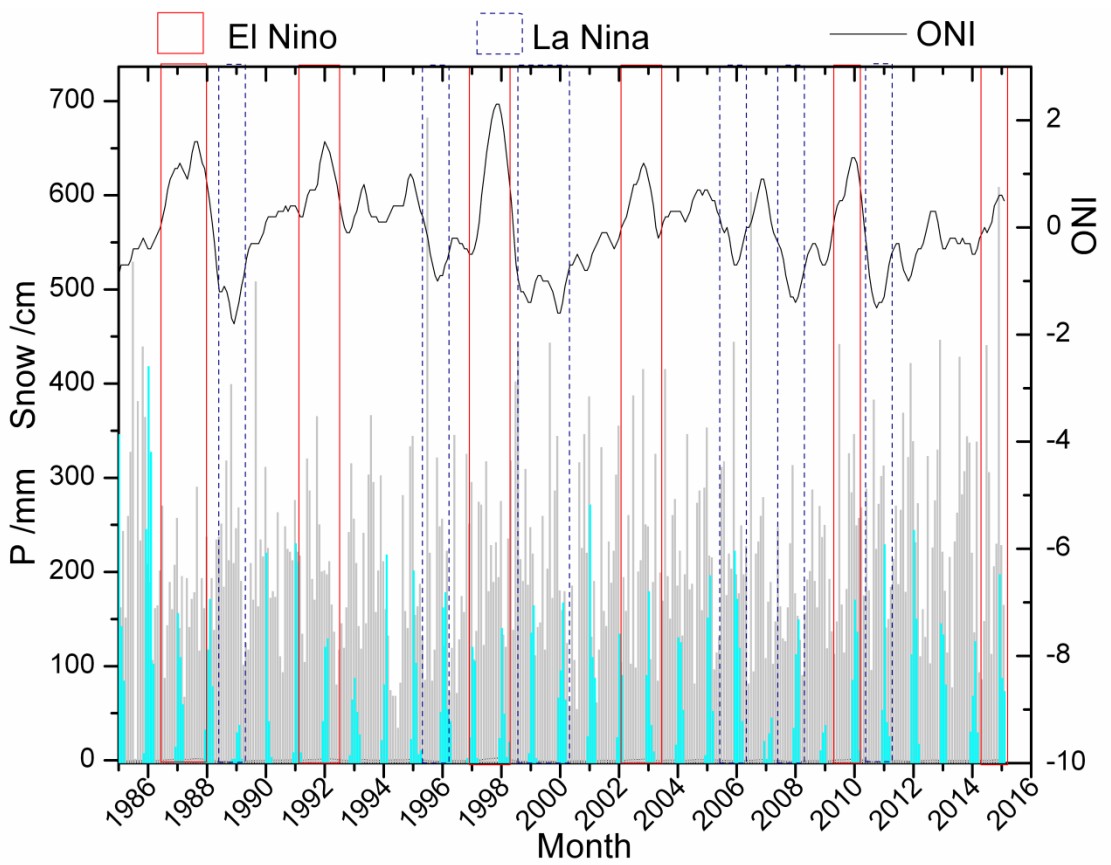

Figure 4 Rainfall (P), snowfall (Snow), oceanic Nino index (ONI), El Niño and La Niña events

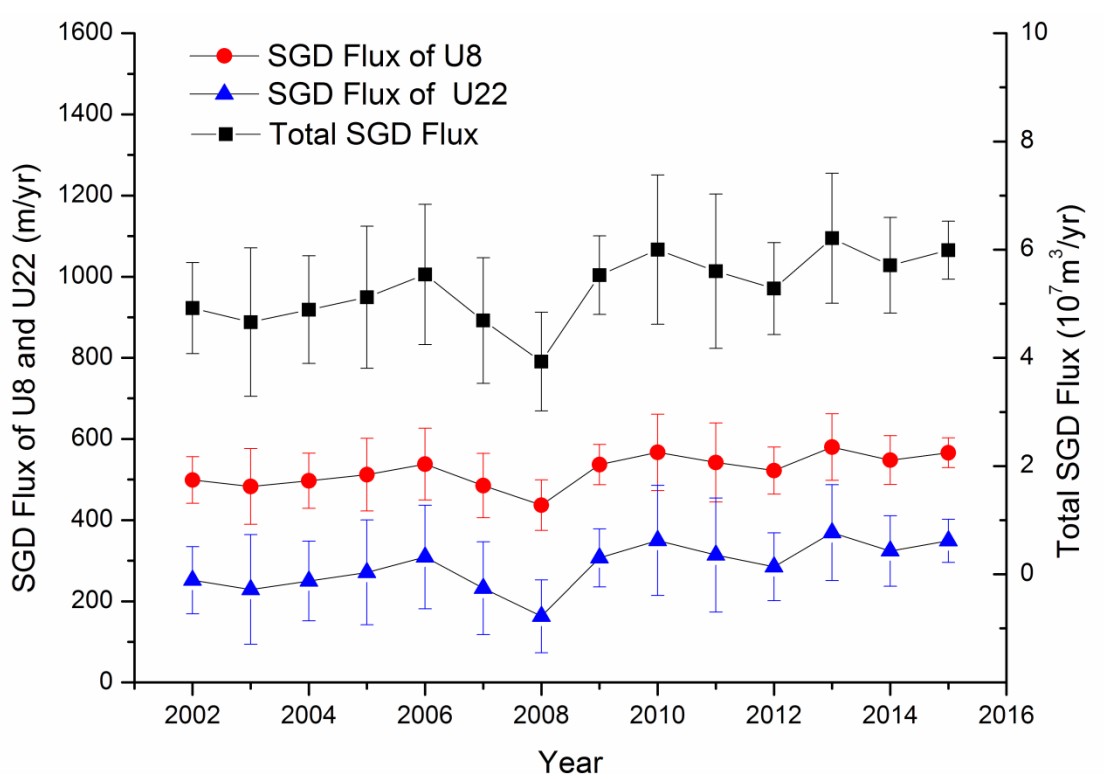

Figure 5 Estimated SGD flux by groundwater table

