# Peer review of "Temporal variations of groundwater table and implications for submarine groundwater discharge: A three-decade case study in Central Japan"

_Hydrology and Earth System Sciences, 2017_

## Referee Comment (RC1) · Anonymous Referee #1 · 17 Mar 2017

The submarine groundwater discharge is likely an important source of nutrients and pollutants to the coastal oceans but is poorly studied. This manuscript provides some long-term data and forecasts the effects of climate change on the SGD. The subject is significant and timely. My major concern is on whether the conclusions are robust. The rain/snow fall varies by 30%. The estimations of ET, river outflow and groundwater discharge are also subject to large uncertainties. The authors should try to propagate the errors and find out what the uncertainty is for the estimated SGD discharge. The readers would then get a better idea whether the forecast is reasonable. A minor issue is that reading numbers in a table does not easily give a trend. A figure replacing Table 3 would do the job more effectively.

---

## Referee Comment (RC2) · Anonymous Referee #1 · 23 Apr 2017

The authors have replied my concerns and I have no further comments.

---

## Author Comment (AC1) · 23 Apr 2017

Thanks for the constructive comments on "Temporal variations of groundwater table and implications for submarine groundwater discharge: A three-decade case study in Central Japan" by Bing Zhang et al.

The responses to the comments are marked blue text as below.

**Reply to Interactive comment of Anonymous Referee #1**

**The submarine groundwater discharge is likely an important source of nutrients and pollutants to the coastal oceans but is poorly studied. This manuscript provides some long-term data and forecasts the effects of climate change on the SGD. The subject is significant and timely.**

**1. My major concern is on whether the conclusions are robust. The rain/snow fall varies by 30%. The estimations of ET, river outflow and groundwater discharge are also subject to large uncertainties. The authors should try to propagate the errors and find out what the uncertainty is for the estimated SGD discharge. The readers would then get a better idea whether the forecast is reasonable.**

Reply: We calculate the variations of water budgets in Table 2 as the comment. The standard deviation of evapotranspiration, river runoff, groundwater usage and SGD were calculated by the variation of precipitation.

Please see the Table 2, page 12 in the revision.

Table 2 Description and prediction of rainfall, snow and water budget

| | Average (Mean±SD) mm/yr | | Water budget* $10^7 \, m^3/yr$ | | | | |
|---|---|---|---|---|---|---|---|
| | rainfall | snowfall | Precipitation | Evapo-transpiration | River runoff | Groundwater usages | SGD |
| 1976~1996 | 2311±616 | 4492±2629 | 47±12.53 | 11±2.93 | 28±7.46 | 2±0.53 | 6±1.6 |
| 1997~2015 | 2652±300 | 3282±1346 | 54±6.11 | 13±1.47 | 32±3.62 | 2±0.23 | 6±0.68 |
| 1976~2015 | 2473±516 | 3850±2110 | 50±10.43 | 12±2.5 | 30±6.26 | 2±0.42 | 6±1.25 |
| 2010~2030 | 2949±150 | 2573±987 | 60±3.05 | 14.4±0.73 | 36±1.83 | 2.4±0.12 | 7.2±0.37 |
| 2030~2050 | 3147±695 | 970±387 | 64±14.13 | 15.36±3.39 | 38.4±8.48 | 2.56±0.57 | 7.68±1.7 |

*Water budget is calculated by percentage of evapotranspiration (24%), river runoff (60%), groundwater usages (4%) and submarine groundwater discharge (SGD, 12%) to precipitation from 1976 to 2015 in Uozu.

Furthermore, the uncertainty of evapotranspiration, and SGD was discussed in section 5.3. The uncertainty was also added in the conclusions.

Please see the lines 19-25 in page 7 in the revision.

According to the terrestrial water budget, an estimated $33\times10^8$ m$^3$/yr of groundwater discharged from the continental shelf into Toyama Bay as fresh submarine groundwater discharge (Ito and Fuji, 1993; Zhang and Satake, 2003). The estimated submarine groundwater discharge is approximately $6\times10^7$ m$^3$/yr, which is 20 % of the river's outflow ($30\times10^7$ m$^3$/yr) in Uozu (Kokusai Kogyo Co. Ltd., 2002) (Table 2). However, the precipitation varies under climate change, because the ratio of rainfall and snow would increase. Furthermore, the variations of meteorological parameters, e.g. temperature, humidity, may cause the changes of evapotranspiration (Cong et al., 2009; Shimizu et al., 2015), as well as river runoff, groundwater discharge in the water budget. Thus, the uncertainty of percentage of evapotranspiration, river runoff and groundwater discharge in the total water budget may exist.

lines 10-15 in page 8 in the revision

(3) SGD flux was controlled by the hydraulic gradient of the coastal groundwater. The linear regression between SGD flux and the groundwater table was established. The historic SGD flux was estimated by groundwater table variation. The upward trend of the precipitation and groundwater table may indicate an increase in SGD flux although with some uncertainty.

**2. A minor issue is that reading numbers in a table does not easily give a trend. A figure replacing Table 3 would do the job more effectively.**

Reply: According to this suggestion, we changed Table 3 to Figure 5 to clarify the trend of SGD flux.

Please see the figure 5 (page 17) in the revision.

[Figure]

Figure 5 Estimated SGD flux by groundwater table

---

## Referee Comment (RC3) · Anonymous Referee #2 · 7 May 2017

This paper delivers important information on the link between climate change and SGD. The approaches are valid, and the manuscript is well written. However, followings should be taken into consideration before this manuscript is accepted for publication in HESS. (1) Relationship between groundwater and SGD: authors cite the paper Zhang et al., 2005 in order to calculate SGD based on a water table. This is very critical part and thus should be clearly explained with respect to methods, assumptions, uncertainties, and limitations. In addition, the results of this approach on SGD can be compared with the salinity data from coastal waters if there are any links (of course, there are many other factors controlling seawater salinities). (2) Implications for SGD: Authors state the importance of SGD on marine productivity and ecosystems. If they

have dissolved inorganic nitrogen (DIN) data in groundwater, authors can strengthen this paper much more. If they do not have those data, they can use some reasonably assumed data to calculate SGD-driven nutrient fluxes and their changes for the last three decades. Then, new production supported by SGD can be inferred from these calculations and state implications on ecosystems. (3) Rounding off problems: authors include many values (42.72 for water table, 14.76 for rain increase, 634.9 mm, and 15.36, 7.68 in Table 2...) throughout the entire manuscript. I think that they cannot measure the values with such accuracies. Please take care of all significant figures. (4) References: Authors should include important original papers and latest papers in references.

---

## Author Comment (AC2) · 24 May 2017

Thanks for the constructive and detailed comments on "Temporal variations of groundwater table and implications for submarine groundwater discharge: A three-decade case study in Central Japan" by Bing Zhang et al.

The responses to the comments are marked blue text as below.

**Reply to Interactive comment of Anonymous Referee #2**

**This paper delivers important information on the link between climate change and SGD. The approaches are valid, and the manuscript is well written. However, followings should be taken into consideration before this manuscript is accepted for publication in HESS.**

**(1) Relationship between groundwater and SGD: authors cite the paper Zhang et al., 2005 in order to calculate SGD based on a water table. This is very critical part and thus should be clearly explained with respect to methods, assumptions, un- certainties, and limitations. In addition, the results of this approach on SGD can be compared with the salinity data from coastal waters if there are any links (of course, there are many other factors controlling seawater salinities).**

Reply: Thanks for your suggestions.

1) We established the relationship between SGD flux and groundwater table by the Darcy's law, because the sand aquifer in Uozu is suitable for using Darcy equation. Also, the method and assumptions were added in the revision; the un-certainties and limitations were also added, please see Table 2.

Please see line 5 to 15 in page 6, some text in the revision is as follow:

    The Darcy's law describes the water flow through a porous medium (sand). The groundwater aquifer in Uozu is sand. …. The SGD flux we estimated is 8 m and 22 m off Uozu. However, the fresh SGD flux at 40 – 100 m is about 2 to 4 times of 0 – 40 m (Hatta and Zhang, 2013). Compared to the results of water budget (Table 2), the fresh SGD maybe underestimated. However, since the groundwater table is easy to determine, the fresh SGD flux could be estimated by the equations 2 and 3 in the coastal sand aquifer.

2) The salinity data is very useful, comparing to seawater salinities. The groundwater salinity is very low. So, the coastal groundwater and SGD is fresh. As your suggestion and advice, we will determine and monitoring the salinity in our future research.

**(2) Implications for SGD: Authors state the importance of SGD on marine productivity and ecosystems. If they have dissolved inorganic nitrogen (DIN) data in groundwater, authors can strengthen this paper much more. If they do not have those data, they can use some reasonably assumed data to calculate SGD-driven nutrient fluxes and their changes for the last three decades. Then, new production supported by SGD can be inferred from these calculations and state implications on ecosystems.**

Reply: Thanks for your suggestion. Since we do not have the long series data of DIN in groundwater, we added average value of DIN ($NO_3^-$) from a reference in the same study area in the revision. The estimated SGD flux is described in Table 2 and Figure 5. The SGD flux increase over three decades, so the DIN flux in the SGD may also elevated these years.

Please see the lines 30 in page 7 in the revision.

The average concentration of $NO_3^-$ in fresh SGD (0.69 mg/L) is larger than riverine input (0.18 mg/L) (Hatta and Zhang, 2013). The estimated SGD flux is described in Table 2 and Figure 5. The SGD flux increase over three decades, the DIN flux in the SGD to Toyama bay may increase.

**(3) Rounding off problems: authors include many values (42.72 for water table, 14.76 for rain increase, 634.9 mm, and 15.36, 7.68 in Table 2 ...) throughout the entire manuscript. I think that they cannot measure the values with such accuracies. Please take care of all significant figures.**

Reply: We checked and revised the rounding off problems throughout the entire manuscript.

The accuracy of groundwater table is 1 cm, after rechecking the original data. The values for groundwater table in text and Table 1, such as 42.72 m, would be correct in the accuracy of 0.01 m.

We corrected the rounding off problems of rainfall, snowfall and water budget in the text and Table 2. The accuracy of rainfall and snow fall is 1mm and 1 cm, respectively. The values of water budget in Table 2 are calculated by percentage to precipitation. These values are rounded to 2 significant figures.

Please see line 27 in page 5, Table 2 in the revision.

There were six El Niño and six La Niña events from 1985 to 2015 (Fig. 4). The seasonal rainfall and snowfall during El Niño events were 642 mm and 155 cm, and those during La Niña events were 635 mm and 157 cm, respectively.

Table 2 Description and prediction of rainfall, snow and water budget

| | Average (Mean±SD) mm/yr | | Water budget* $10^7$ m$^3$/yr | | | | |
| --- | --- | --- | --- | --- | --- | --- | --- |
| | rainfall | snowfall | Precipitation | Evapo-transpiration | River runoff | Groundwater usages | SGD |
| 1976~1996 | 2311±616 | 4492±2629 | 47±13 | 11±2.9 | 28±7.5 | 2.0±0.5 | 6.0±1.6 |
| 1997~2015 | 2652±300 | 3282±1346 | 54±6.1 | 13±1.5 | 32±3.6 | 2.0±0.2 | 6.0±0.68 |
| 1976~2015 | 2473±516 | 3850±2110 | 50±10 | 12±2.5 | 30±6.3 | 2.0±0.4 | 6.0±1.2 |
| 2010~2030 | 2949±150 | 2573±987 | 60±3.1 | 14±0.73 | 36±1.8 | 2.4±0.1 | 7.2±0.37 |
| 2030~2050 | 3147±695 | 970±387 | 64±14 | 15±3.4 | 38±8.5 | 2.6±0.6 | 7.7±1.7 |

*Water budget is calculated by percentage of evapotranspiration (24%), river runoff (60%), groundwater usages (4%) and submarine groundwater discharge (SGD, 12%) to precipitation from 1976 to 2015 in Uozu.

**(4) References: Authors should include important original papers and latest papers in references.**

Reply: We added important original papers and latest papers in references.

Please see lines 15, 30 in page6; lines 2, 30 in page 7 ; lines 42 in page 19; lines 13, 27 in page 20 in the revision.

Thanks again for carefully, scientifically and detailed review and critical suggestions.

---

## Referee Comment (RC4) · Anonymous Referee #2 · 25 May 2017

I think that authors took into account most of my major comments in the revised version. I do not have further comments.

---

## Referee Comment (RC5) · Anonymous Referee #1 · 25 May 2017

The authors have answered my concerns successfully and I have no further comments.